# Pyrazole-Based Water-Soluble Dendrimer Nanoparticles as a Potential New Agent against Staphylococci

**DOI:** 10.3390/biomedicines10010017

**Published:** 2021-12-23

**Authors:** Silvana Alfei, Chiara Brullo, Debora Caviglia, Gabriella Piatti, Alessia Zorzoli, Danilo Marimpietri, Guendalina Zuccari, Anna Maria Schito

**Affiliations:** 1Department of Pharmacy (DIFAR), University of Genoa, Viale Cembrano, 16148 Genoa, Italy; brullo@difar.unige.it (C.B.); zuccari@difar.unige.it (G.Z.); 2Department of Surgical Sciences and Integrated Diagnostics (DISC), University of Genoa, Viale Benedetto XV, 6, 16132 Genoa, Italy; debora.caviglia@edu.unige.it (D.C.); gabriella.piatti@unige.it (G.P.); amschito@unige.it (A.M.S.); 3Stem Cell Laboratory and Cell Therapy Center, IRCCS Istituto Giannina Gaslini, Via Gerolamo Gaslini 5, 16147 Genoa, Italy; alessiazorzoli@gaslini.org (A.Z.); danilomarimpietri@gaslini.org (D.M.)

**Keywords:** fourth-generation, polyester-based, lysine-modified dendrimer, physical encapsulation, 3,5-diphenyl-pyrazole derivative (BBB4), Gram-positive MDR isolates, *Staphylococcus* genus, MICs, cytotoxicity on human keratinocytes, selectivity index

## Abstract

Although the antimicrobial potency of the pyrazole nucleus is widely reported, the antimicrobial effects of the 2-(4-bromo-3,5-diphenyl-pyrazol-1-yl)-ethanol (BBB4), found to be active against several other conditions, have never been investigated. Considering the worldwide need for new antimicrobial agents, we thought it noteworthy to assess the minimum inhibitory concentration (MICs) of BBB4 but, due to its scarce water-solubility, unequivocal determinations were tricky. To obtain more reliable MICs and to obtain a substance also potentially applicable in vivo, we recently prepared water-soluble, BBB4-loaded dendrimer nanoparticles (BBB4-G4K NPs), which proved to have physicochemical properties suitable for clinical application. Here, with the aim of developing a new antibacterial agent based on BBB4, the BBB4-G4K NPs were tested on several strains of different species of the *Staphylococcus* genus. Very low MICs (1.5–3.0 µM), 15.5–124.3-fold lower than those of the free BBB4, were observed against several isolates of *S. aureus* and *S. epidermidis*, the most pathogenic species of this genus, regardless of their resistance patterns to antibiotics. Aiming at hypothesizing a clinical use of BBB4-G4K NPs for staphylococcal skin infections, cytotoxicity experiments on human keratinocytes were performed; it was found that the nano-manipulated BBB4 released from BBB4-G4K NPs (*LD*_50_ 138.6 µM) was 2.5-fold less cytotoxic than the untreated BBB4 (55.9 µM). Due to its physicochemical and biological properties, BBB4-G4K NPs could be considered as a promising novel therapeutic option against the very frequent staphylococcal skin infections.

## 1. Introduction

Over the past twenty years, the number of multidrug-resistant bacteria (MDR) has grown exponentially [1]. MDR pathogens are bacteria resistant to at least three or more classes of antimicrobial drugs [2]; their prevalence has been increasing, especially in recent years [3], posing serious risks to public health.

MDR bacteria belong to both Gram-positive, Gram-negative, and Mycobacteria groups, and they are all indifferent to the inhibitory action of antibiotics, though in different ways. To compensate for the therapeutic failures that often result from this condition, the amount of antibiotics consumed has, paradoxically, significantly increased [4]. As a result, there has been a notable deterioration in the rates of antimicrobial resistance among pathogens, due to which it is now crucial to identify new active drugs [5]. 

Although pathogens of Gram-negative species are considered especially clinically relevant superbugs, which are capable of worrying levels of resistance and are responsible for numerous therapeutic failures [1], antibiotic resistance has also become a great obstacle in treating infections caused by many Gram-positive bacteria. Today, very often, MDR Gram-positive pathogens, such as methicillin-resistant *Staphylococcus aureus* and *Staphylococcus epidermidis* (MRSA and MRSE, respectively), penicillin-resistant *Streptococcus pneumoniae*, *Enterococcus faecium*, and vancomycin-resistant *E. faecalis* (VRE), frequently become tolerant to currently available agents, thus requiring innovative therapeutic strategies and/or the development of new drugs [6]. 

They are responsible for infections against which traditional antibiotics are no longer effective, thus causing an increasing number of deaths in hospitals, long-term care facilities, and community settings. The scrutiny of antibiotic resistance is a global health concern [7,8,9,10,11,12], and the search for new antibacterial agents, acting by means of mechanisms different from those of available antibiotics, and having a lower propensity to develop resistance, represents one of the greatest challenges for researchers [13]. 

An innovative therapeutic strategy is the bacteriophage therapy, or “phage therapy”, which uses bacteria-specific viruses, which are engineered to only infect a certain bacteria species and to kill antibiotic-resistant bacteria [14]. However, one major drawback to phage therapy is the evolution of phage-resistant microbes, as was seen in most of the phage therapy experiments aimed to treat sepsis and intestinal infection [15]. 

Concerning the development of new antibacterial drugs, active where common antibiotics no longer work, the five-membered heterocyclic diazole ring of pyrazole has been shown to have numerous biological activities, including antimicrobial effects [16,17]. The pyrazole derivatives are mainly of synthetic derivation, and only a few molecules containing the pyrazole nucleus are of natural origin. In this regard, the first example of a natural product containing the pyrazole ring, endowed with anti-diabetic activity, was the amino acid *L*-α-amino-β-(pyrazolyl-*N*)-propanoic acid [(S)-β-pyrazolyl alanine], isolated in 1957 from the *Citrullus vulgarisin* juice of watermelon [18]. The pyrazole nucleus possesses almost all kinds of pharmacological activities, but, while several of its medicinal properties have been reported since the year 1944, the first studies reporting the antimicrobial effects of the pyrazole derivatives have been published much more recently, particularly after the year 2000 [19,20]. Consequently, the 2-(4-bromo-3,5-diphenyl-pyrazol-1-yl)-ethanol (BBB4, Figure 1), synthetized for the first time by Bondavalli et al. and found to be active as an analgesic, hypotensive, anti-inflammatory, local anaesthetic, inhibitor of motor activity in mice and rats, and as an antiplatelet aggregative agent in vitro in 1998 [16], has never been studied for its possible antimicrobial properties.

Nowadays, the antimicrobial properties of molecules containing the pyrazole ring are extensively documented [21], and frequently, they are attributed to the amino groups present in the structure [22]. Therefore, aiming at meeting the global need for new therapeutic options against infections by MDR bacteria, we thought that BBB4, until now not yet investigated precisely for this ability, could be a suitable candidate for the development of a novel antibacterial agent.

To determine the antibacterial effects of BBB4, we first determined the MICs of BBB4 against representative strains of Gram-positive and Gram-negative species. Unfortunately, due to its insolubility in water, even if a tendency of BBB4 to be active has been highlighted, particularly against Gram-positive species, the experiments carried out have provided results that are difficult to unambiguously interpret. Moreover, despite being active, a possible clinical application of the insoluble free form of BBB4 would remain utopian unless water-soluble BBB4 formulations were developed. To address the problems related to the solubility of bioactive compounds, several nanotechnological strategies have provided very promising results [23,24,25]. In this field, due to their nonpareil physicochemical properties, dendrimers are extensively applicated [24,26,27,28,29,30]. Accordingly, using a dendrimer found to be non-cytotoxic for HeLa cells as a solubilizing agent, we have recently prepared water-soluble BBB4-loaded nanoparticles (NPs) (BBB4-G4K NPs) with improved physicochemical properties, suitable both for performing reliable antimicrobial essays and for a future clinical application [16]. 

Here, the obtained BBB4-G4K NPs were evaluated for their effects on both the bacterial and the normal eucaryotic cells. Particularly, a preliminary screening was re-performed, which showed a remarkable and selective antibacterial activity against the *Staphylococcus* genus. Therefore, we have studied in detail the effects of BBB4-G4K NPs on several clinical isolates of staphylococci of different species, obtaining excellent results. 

Finally, to evaluate the feasibility of the clinical application of BBB4-G4K for the treatment of skin infections, very frequently caused by MRSA and MRSE isolates [31], the cytotoxicity of BBB4-G4K NPs on human keratinocytes was evaluated. In parallel, G4K and BBB4 were also tested under the same conditions for comparative purposes. 

### 1.1. Insight into the Staphylococcus Genus 

As for staphylococci, they are a group of bacteria that include more than 30 species. Many of these are pathogenic to humans and can cause different types of infections, including skin infections, which are the ones they most commonly cause [31], bacteraemia, bone infections, endocarditis, food poisoning, pneumonia, and toxic shock syndrome (TSS), a life-threatening condition caused by toxins. However, one of the species of the *Staphylococcus* genus is particularly fearsome, namely *S. aureus*, a round-shaped bacterium that can belong to the human microbiota, especially in the anterior nostrils, on the skin, and in the lower reproductive tract of women. It has been estimated that 20% to 30% of the human population are long-term carriers of *S. aureus* [32,33,34,35,36]. Although *S. aureus* usually acts as a commensal of the human microbiota, it can also become an opportunistic pathogen [37]. As with the most dangerous members of *Staphylococcus* genus, *S. aureus* frequently causes skin infections, such as pimples [37], impetigo, boils, cellulitis, folliculitis, carbuncles, scalded skin syndrome, and abscesses. Additionally, *S. aureus* can be responsible for respiratory infections, such as sinusitis, or food intoxications. *S. aureus* can also cause more severe and life-threatening diseases, such as pneumonia, meningitis, osteomyelitis, endocarditis, toxic shock syndrome, bacteraemia, and sepsis. *S. aureus* potency depends on its capability to produce virulence factors such as protein toxins and to express a cell-surface protein that binds and inactivates antibodies. It is still one of the five most common causes of hospital-acquired infections and is often the cause of wound infections following surgery. Each year, about 500,000 patients in the hospitals of the United States contract a staphylococcal infection by *S. aureus* [38]. Up to 50,000 deaths each year in the USA are due to *S. aureus* infections [39]. *S. aureus* infections are usually treated with penicillinase-resistant beta-lactams, but due to the high tendency of the specie to develop resistance to such antibiotics (as is the case with MRSA strains), vancomycin or other newer antibiotics may be required. Some strains are partially or totally resistant to all but the newest antibiotics, which include linezolid, tedizolid, quinupristin/dalfopristin, daptomycin, telavancin, dalbavancin, oritavancin, tigecycline, eravacycline, omadacycline, delafloxacin, ceftobiprole (not available in the USA), ceftaroline, and lefamulin [31], but cases of resistance towards some of these antibiotics are also increasingly reported. In particular, in the following Section 1.1.1, an overview concerning the current treatment options to counteract MRSA is reported, and the relevance of the results reported herein is also highlighted.

#### 1.1.1. Current Treatment Options against *S. aureus* and the Relevance of This Study

MRSA is a particular isolate of *S. aureus*, distinct from other strains of the same species, which cannot be inhibited by methicillin and/or oxacillin. MRSA has developed (through natural selection) or acquired (through horizontal gene transfer) a multiple drug resistance to β-lactam antibiotics, which include some penicillin derivatives, such as methicillin and oxacillin, and the cephalosporins [40]. MRSA is common in hospitals, prisons, and nursing homes, where there are in-patient immunocompromised people with open wounds and/or invasive devices such as catheters, who are at greater risk of hospital-acquired infections. Although MRSA began as a hospital-acquired infection, it has become community-acquired, as well as livestock-acquired. Specific terms exist to indicate healthcare-associated or hospital-acquired MRSA (HA-MRSA), community-associated MRSA (CA-MRSA), and livestock-associated MRSA (LA-MRSA). Against MRSA, the use of concurrent treatment with vancomycin or other β-lactam agents may be successful thanks to a synergistic effect [41]. CA-MRSA has a greater spectrum of antimicrobial susceptibility to sulfa antibiotics such as co-trimoxazole (trimethoprim/sulfamethoxazole), tetracyclines (doxycycline and minocycline), and clindamycin (for osteomyelitis) [42]. In soft-tissue infections, MRSA can be eradicated with linezolid [43], which is successful in 87% of people [43], while vancomycin is successful in approximately 49% of people [40]. For treating patients with MRSA pneumonia, the Infectious Disease Society of America recommends vancomycin, linezolid, or clindamycin [44]. Additionally, vancomycin and teicoplanin are glycopeptide antibiotics used to treat MRSA infections [45], even though treatment of MRSA infection with vancomycin can be complicated, due to its inconvenient route of administration. Moreover, the efficacy of vancomycin against MRSA is inferior to that of anti-staphylococcal β-lactam antibiotics against methicillin-susceptible *S. aureus* (MSSA) [46,47].

Additionally, several newly discovered strains of MRSA have shown antibiotic resistance, even to vancomycin and teicoplanin. Particularly, strains with intermediate (4–8 μg/mL) levels of resistance began appearing in the late 1990s, while the first documented strain with complete (>16 μg/mL) resistance to vancomycin, termed vancomycin-resistant *S. aureus* (VRSA), appeared in the United States in 2002 [48]. 

Oxazolidinones such as linezolid became available in the 1990s and are comparable to vancomycin in effectiveness against MRSA. However, linezolid resistance in *S. aureus* was already reported in 2001 [49]. More information is needed to determine the effectiveness of specific antibiotics in the therapy of surgical site infections (SSIs) by MRSA [44], and it is not clear (high risk of bias) that linezolid could be better than vancomycin for the eradication of MRSA SSIs [40]. The same goes for MRSA colonization in nonsurgical wounds such as traumatic wounds, burns, and chronic ulcers (i.e., the diabetic ulcer, pressure ulcer, arterial insufficiency ulcer, and venous ulcer), for which no conclusive evidence has been found about the best antibiotic regimen to treat the MRSA colonization [50]. 

In this alarming scenario, made of missing epidemiologic evidence, a plethora of uncertainties, also due to the interindividual responses of patients to existing antibiotics, and of a decreasing efficacy of available drugs, the development of new curative options against MRSA infections is urgent. Consequentially, the overall merit of the present study consists of having identified a new powerful antibacterial agent based on nanotechnologically modified BBB4 (BBB4-G4K NPs), which, thanks to its previously demonstrated [16] physicochemical characteristics, could be advisable for biomedical applications. Such NPs have proven to be selectively active against different species of the *Staphylococcus* genus, and to be able to inhibit their growth whatever their resistance to methicillin or their resistance to existing antibiotics, amoxicillin and linezolid included. As a result, BBB4-G4K NPs may represent a possible new therapeutic option for addressing the severe staphylococcal infections caused by MDR species, which are of increasing concern around the world. 

## 2. Materials and Methods

### 2.1. Chemicals Substances and Instruments

The biodegradable cationic dendrimer NPs loaded with BBB4 (BBB4-G4K NPs) used in this study were recently synthesized according to the synthetic procedure reported by Alfei and collaborators [16]. The experimental details and characterization data are available in the Appendix A. In addition, the dose-dependent cytotoxicity experiments performed with G4K on eukaryotic ovarian cancer cells (HeLa) are available in Appendix A.

Details concerning the chemical materials and methods, as well as the instruments used for the physicochemical and biological characterization of BBB4-G4K NPs, are consultable in our previous work [16].

### 2.2. Microbiological Studies

#### 2.2.1. Bacterial Strains Employed in This Study

A total of 28 strains belonging to the Gram-positive and Gram-negative species were used in this study. Except for an ATCC *S. aureus* strain (*S. aureus* ATCC 29213), all the bacteria were clinical isolates, belonging to a collection obtained from the School of Medicine and Pharmacy of the University of Genoa (Italy). They were identified by VITEK^®^ 2 (Biomerieux, Firenze, Italy) or the matrix-assisted laser desorption/ionization time-of-flight (MALDI-TOF) mass spectrometric technique (Biomerieux, Firenze, Italy). Particularly, two strains were of Gram-negative species (1 *Escherichia coli* and 1 MDR *Pseudomonas aeruginosa*). Twenty-six strains were Gram-positive, including two isolates of the genus *Enterococcus* (1 *E. faecalis* and 1 *E. faecium*), one sporogenic *B. subtilis*, and 23 clinical isolates of the genus *Staphylococcus*, including one ATCC *S. aureus*, 1 methicillin-susceptible (MSSA) *S. aureus*, 9 methicillin-resistant (MRSA) *S. aureus*, and 6 methicillin-resistant (MRSE) *S. epidermidis*, including 2 isolates also resistant to linezolid, 1 *S. haemoliticus*, 1 *S. hominis*, 1 *S. lugdunensis*, 1 *S. saprophyticus*, 1 *S. symulans*, and 1 *S. warneri*.

#### 2.2.2. Determination of the Minimal Inhibitory Concentrations (MICs) 

To investigate the antimicrobial activity of the substances considered in this study, the MICs were determined by following the microdilution procedures detailed by the European Committee on Antimicrobial Susceptibility Testing EUCAST [51]. 

Briefly, overnight cultures of bacteria were diluted to yield a standardized inoculum of 1.5 × 10^8^ CFU/mL. Aliquots of each suspension were added to 96-well microplates containing the same volumes of serial 2-fold dilutions (ranging from 1 to 128 μg/mL) of BBB4 and BBB4-G4K to yield a final concentration of about 5 × 10^5^ cells/mL. The plates were then incubated at 37 °C. After 24 h of incubation at 37 °C, the lowest concentration of the substances that prevented a visible growth was recorded as the MIC. All MICs were obtained in triplicate, the degree of concordance in all the experiments was 3/3 and the standard deviation (±SD) was zero. 

### 2.3. Evaluation of BBB4, G4K, and BBB4-G4K NPs Cytotoxicity on Human Keratinocytes 

#### 2.3.1. Experimental Protocol for Cell Culture

Human skin keratinocytes cells (HaCaT), obtained thanks to a generous gift from the Laboratory of Experimental Therapies in Oncology, IRCCS Istituto Giannina Gaslini (Genoa, Italy), were grown as a monolayer in RPMI 1640 medium supplemented with 10% fetal bovine serum (*v*/*v*), 1% penicillin-streptomycin, and 1% glutamine (Euroclone S.p.A., Milan, Italy), cultured in T-25 cm^2^ plastic flasks (Corning, NY, USA) and maintained at 37 °C in a 5% CO_2_ humidified atmosphere. Cells were tested and characterized at the time of experimentation, as previously described [52].

#### 2.3.2. Viability Assay

HaCaT cells were seeded in 96-well plates (at 4 × 10^3^ cells/well) in complete medium and cultured for 24 h. The seeding medium was removed and replaced with fresh complete medium that had been supplemented with increasing concentrations of empty dendrimer (G4K), BBB4 or BBB4-G4K (0 μM, 1 μM, 5 μM, 10 μM, 15 μM, 20 μM, 25 μM, 50 μM, 75 μM, and 100 μM). The cells (quadruplicate samples for each condition) were then incubated for an additional 4, 12 or 24 h. The effect on cell growth was evaluated by the fluorescence-based proliferation and cytotoxicity assay CyQUANT^®^ Direct Cell Proliferation Assay (Thermo Fisher Scientific, Life Technologies, Monza Brianza, Italy), according to the manufacturer’s instructions. Briefly, at the selected times an equal volume of detection reagent was added to the cells in culture and incubated for 60 min at 37 °C. The fluorescence of the samples was measured using the mono-chromator-based M200 plate reader (Tecan, Männedorf, Switzerland), set at 480/535 nm. The experiments were carried out at least three times and samples were run in quadruplicate.

### 2.4. Statistical Analyses

Concerning the cytotoxicity studies, the statistical significance of differences between the experimental and control groups and between the different samples was determined via a two-way analysis of variance (ANOVA) with the Bonferroni correction. The analyses were performed with Prism 5 software (GraphPad, La Jolla, CA, USA). Asterisks indicate the following *p*-value ranges: * = *p* < 0.05, ** = *p* < 0.01, *** = *p* < 0.001. Concerning the *MIC* values, experiments were made in triplicate and the concordance degree was 3/3 and the ±SD was zero.

## 3. Results and Discussion

### 3.1. A Quick Look at the Main Characteristics of BBB4-G4K NPs

The main characteristics of BBB4-G4K NPs have been included in the following Table 1.

### 3.2. Antibacterial Properties of BBB4 and BBB4-G4K NPs

Most of the pharmacological activities of the pyrazole nucleus have been studied since the year 1944, but the first studies on the antimicrobial effects of pyrazole derivatives began after the year 2000 [19,20,21,22]. Particularly, 2-(4-bromo-3,5-diphenyl-pyrazol-1-yl)-ethanol (BBB4), previously shown to be active as an analgesic, hypotensive, anti-inflammatory, local anaesthetic, and inhibitor of motor activity in mice and rats and as an antiplatelet aggregative agent in vitro [16], has never been evaluated for its antimicrobial properties prior to the present study. 

Nowadays, the antimicrobial properties of pyrazole derivatives are extensively documented [21,22]. Imagining that BBB4 could also be a good candidate as a new antibacterial agent, we examined its antibacterial activity on the strains, including the MDR, representative of species both Gram-positive and Gram-negative. To this end, we determined the MICs of BBB4 on an *S. aureus* ATCC strain, on isolates of different species of the *Staphylococcus* genus, on two isolates of the *Enterococcus* genus, and on isolates of *E. coli, Pseudomonas aeruginosa* and *Bacillus subtilis*. Due to the insignificant water solubility of BBB4, the MICs results were difficult to interpret in a non-equivocal way and, as they were not entirely clear, it was only possible to roughly detect the antibacterial potency of BBB4 (Table 2). 

Notably, as the *MIC* value of 128 µg/mL was assumed as the cut-off value, above which the compound was considered inactive, BBB4 was considered inactive against isolates of Gram-negative species and of the *Enterococcus* genus (MICs > 128 µg/mL), poorly active against *B. subtilis* (MICs = 128 µg/mL), and promising against some isolates of the *Staphylococcus* genus, mainly towards strains of the species *S. aureus* and *S. epidermidis* (MICs = 32 µg/mL). 

Consequently, we thought it interesting to deepen the study of the antibacterial activity of BBB4 against a wider range of isolates of the *Staphylococcus* genus. On the other hand, to obtain not-questionable *MIC* values, a strategy to obtain water-soluble BBB4 formulations was necessary. To this end, we recently encapsulated BBB4 in the G4K dendrimer, proven to be devoid of antibacterial properties (results not reported), obtaining water-soluble BBB4-loaded NPs [16] with physicochemical properties also suitable for a future biomedical application. In the current part of the study, we evaluated whether, after this nano-manipulation, the antibacterial properties of the BBB4 already observed had been preserved and/or modified. 

#### MICs of BBB4-G4K NPs

Water-soluble BBB4-G4K NPs were then tested against several isolates of different species of the *Staphylococcus* genus, and the *MIC* values obtained are given in Table 3. 

According to the results reported in Table 3, the BBB4-G4K NPs demonstrated excellent antibacterial activity against all clinical isolates of the *Staphylococcus* genus tested and particularly against the isolates of *S. aureus* and *S. epidermidis* (MICs = 1.5–3.0 µM), including the MRSA and MRSE strains, and the MRSE isolates were also resistant to linezolid. Interestingly, the BBB4-G4K NPs proved to be much more potent than the pristine BBB4, reporting *MIC* values 15.5–62.1-fold and 31.1-fold lower than those of the untreated BBB4 on the MRSA strains and *S. epidermidis* strains resistant to both oxacillin and linezolid, respectively. In addition, the BBB4-G4K NPs displayed MICs 124.3-fold, 31.1-fold, and 62.2-fold lower than those displayed by the untreated BBB4 on *S. lugdunensis*, *S. saprophyticus*, and *S. warneri*, respectively. Differently from the pristine BBB4, the BBB4-G4K NPs were also active on the oxacillin-resistant *S. haemoliticus, S. hominis*, and *S. symulans* strains. Interestingly, if we consider the maximum concentration of BBB4 released from the BBB4-G4K NPs to which bacteria can be exposed (according to the MICs observed for BBB4-G4K NPs, their DL%, and their release profile) (fourth column in Table 3), after nanotechnological modification the antibacterial potency of untreated BBB4 was improved. Particularly, when tested against *S. aureus* (including MRSA), the nanoengineered BBB4, showing MICs of 25.7–51.3 µM, except for strain 18, displayed MICs 1.8–3.6-fold higher than those observed for the untreated BBB4. Similarly, when tested against *S. epidermidis* MRSE (including the MRSE also resistant to linezolid), the nano-modified BBB4 showed MICs 1.8-fold lower than those of the pristine BBB4. Finally, when tested against *S. lugdunensis, S. saprofiticus* and, *S. warneri*, showing MICs of 51.3 µM against all species, the nano-manipulated BBB4 displayed MICs 7.2-, 1.8-, and 3.6-fold lower than those of the untreated BBB4, respectively. Furthermore, pristine water insoluble BBB4, in hypothetical in vivo experiments, would be difficult to administer without the use of harmful solvents and co-solvents, and its consequent poor bioavailability could further adversely affect its efficacy in vivo. Conversely, nanoengineered water-soluble BBB4 in the form of BBB4-G4K NPs could be easily administered and therefore be suitable for biomedical applications. It is interesting to note that, while there are numerous articles on the in vitro antimicrobial activity of non-formulated pyrazole derivatives, those on the in vitro antimicrobial activity of dendrimer-based nano-formulations loaded with molecules having the pyrazole nucleus are completely missing. To our knowledge, there are only two studies in the literature on the nano-encapsulation of two pyrazole derivatives [22,53], but only one concerns the application of nanotechnology to improve the water solubility and the biological anti-tumor activity of a pyrazole derivative [53]. Even though the pyrazole-based nano-formulations reported in the other study were prepared with antibacterial purposes, the nanotechnological modification was concerned with the encapsulation of 1-phenyl pyrazole-3, 5-diamine, 4-[2-(4-methylphenyl) diazenyl] (compound **1**) and 1H- pyrazole-3, 5-diamine, 4-[2-(4-methylphenyl) diazenyl] (compound **2**) into liposomal chitosan emulsions for textile finishing [22]. Therefore, the present study is the first that concerns the successful evaluation of the antibacterial activity of a water-soluble pyrazole-based formulation obtained by a dendrimer-based nanotechnological approach. Moreover, even if it is difficult to make a direct comparison between the results concerning the bacterial growth inhibition activity of BBB4-G4K NPs and those reported for the samples of cotton fabric treated with the chitosan liposomal emulsions of pyrazole compounds previously reported [22], because only the cotton fabric treated with the emulsion at concentration 1.25 mg/mL (1250 µg/mL) inhibited the growth of *S. aureus* ATCC 25923, the BBB4-G4K NPs were 19.5–39-fold more potent, including when tested against MRSA species.

### 3.3. Cytotoxicity of G4K, BBB4, and BBB4-G4K NPs on HaCaT Human Keratinocytes Cells

The solubility in water and a sufficiently high value of the selectivity index (*SI*) are pivotal requirements to make a new molecule or macromolecules worthy of consideration as new therapeutic agent against bacterial infections. The pristine BBB4, during early antimicrobial investigations, had shown the crucial problem of a very poor water solubility. Consequently, in addition to hamper the obtainment of reliable and clear results in current microbiological experimentation in vitro, its insignificant solubility in water would have made almost impossible the administration of BBB4 and limited its efficacy, in a hypothetical future in vivo experimentation. We have solved the drawbacks of BBB4 related to water solubility in our recently published previous work, thus satisfying one of the first important needs [16]. In this work, assessing the remarkable and improved antibacterial properties of BBB4-G4K NPs, we first ascertained the values of *SI* of both BBB4 and the BBB4-G4K NPs, which is the second major requisite for a new drug. Secondly, we established which substance was the most suitable for being developed as a new antibacterial agent against staphylococci. To have an acceptable value of *SI*, a new antimicrobial agent should have low *MIC* values on bacteria, associated with a high lethal dose (expressed as *LD*_50_, which is the dose needed to kill 50% of the cells) on eukaryotic cells. In microbiology, the *SI* value is given by the equation in Equation (1) and provides a measure of the selectivity of the antimicrobial agent for bacteria.
(1)$$SI=LD_{50}/MIC$$

We performed dose and time dependent cytotoxicity experiments on human keratinocytes (HaCaT), and the results from the dose-dependent cytotoxicity experiments performed for 24 h were used to compute the *LD*_50_. The obtained *LD*_50_ and the *MIC* values were used to calculate the *SI* value of BBB4-G4K against each isolate of the *Staphylococcus* genus considered in this study. In parallel, the *SI* values of untreated BBB4 were calculated for comparisons.

#### Dose- and Time-Dependent Cytotoxicity Experiments

Dose- and time-dependent cytotoxicity experiments were performed to evaluate the effects of BBB4-G4K NPs on HaCaT keratinocytes cells. Cytotoxicity experiments under the same conditions were also conducted for BBB4 and G4K to evaluate the reciprocal effects on the original cytotoxicity of the pristine BBB4 and the empty dendrimer. Such experiments were performed on HaCaT keratinocytes mainly because 75% of MRSA infections are localized to skin and soft tissue [31,43]. Consequently, to assess the cytotoxicity of BBB4-G4K NPs, we selected human keratinocytes, which are the primary type of cell found in the epidermis, the outermost layer of the skin, and are more susceptible to colonization by bacteria, fungi, and parasites. The cytotoxic activity of the samples, as a function of their concentrations (1–100 µM), was determined after 4, 12, and 24 h of exposure of the cells. The results are reported in Figure 2a–c. 

As can be seen in Figure 2, for all the compounds the cytotoxic effects were both time- and dose-dependent. Let us first consider the two ingredients, G4K and BBB4. As expected, the cytotoxic profile of the empty dendrimer G4K conformed to that observed by us previously after both 4, 12, and 24 h [54]. Particularly, after 4 h of exposure, at concentrations of 5–100 µM, the G4K was the least toxic compound, with no significative difference in cell viability compared to the control, even at the maximum concentration tested (100 µM, cell viability 97.2%). The G4K was slightly more cytotoxic after 12 h of incubation at 1–25 µM concentrations, while its cytotoxicity improved significantly for higher concentrations, but the cell viability remained >50%, even at 100 µM concentrations (53.3%). After 24 h of exposure, the cytotoxic effects of G4K increased further, and a significant difference in cell viability compared to the control was observed even at the minimum concentration tested (1 µM). Cell viability was maintained at >50% to a concentration of 25 µM (59.3%), while it decreased dramatically for higher concentrations, reaching the 23.1% at 100 µM. The pyrazole derivative BBB4 exerted cytotoxic effects comparable to those of G4K, or slightly higher, resulting in non-cytotoxicity up to a concentration of 15 µM after 4 h of exposure. Its cytotoxicity increased progressively at 20, 25, and 50 µM (viability cells of 90.9, 86.4, and 70.5%, respectively), while at over 50 µM it remained unchanged. After 12 h of exposure, the BBB4, while slightly more cytotoxic than the G4K up to a concentration of 10 µM, at higher concentrations it displayed a cytotoxicity comparable to that of G4K (15–50 µM), or significantly lower (75–100 µM). Notably, at a BBB4 100 µM concentration, the cell viability was 66.9%, while that observed for the G4K was 53.3%. After 24 h of cell exposure to BBB4, the detected cytotoxicity was lower than that observed for G4K at all the concentrations tested, but the cell viability was <50% from the concentration of 50 µM (45.9%). Moreover, as observed after 12 h of exposure, the cell viability did not change significantly for higher BBB4 concentrations. As for the BBB4-G4K NPs, the results obtained were very different from those observed for the NPs made of the same dendrimer G4K but loaded with ursolic acid (UA-G4 NPs) and those of the recently published [54]. In that study, the encapsulation of UA in G4K led to a reduction in the intrinsic cytotoxicity of the two ingredients, resulting in UA-loaded NPs significantly less cytotoxic than the two compounds when administered alone [54]. On the contrary, in this case, after 4, 12, and 24 h of exposure, and depending on the time of the experiments, a dramatic improvement in the cytotoxic effects of the BBB4-G4K NPs was observable starting from the 15 µM concentration, resulting in a reduction in cell viability below 50% [cell viability of 38.0% (4 h), 16.5% (12 h), and 13.7% (24 h)]. For higher concentrations of BBB4-G4K NPs, the cell viability remained essentially unchanged. For lower concentrations, the BBB4-G4K NPs exerted a cytotoxicity comparable to or lower than that of the G4K and BBB4 after 4 h of exposure, comparable (at 1 µM concentration) or increasingly higher after 12 h, and always higher after 24 h of exposure. Overall, it might seem that this time the nanotechnological manipulation of BBB4, with the same G4K dendrimer previously employed with totally different results, has led to a substantial improvement in the cytotoxicity of the two ingredients. In effect, this early deduction is incorrect as the high cytotoxicity observed for the BBB4-G4K NPS is not due to the nano-formulation but to the high amount of BBB4 that the BBB4-G4K NPs contain (28.8%) and that the BBB4-G4K NPs release (95.5%) according to their DL% and release profile [16]. Practically, the HaCaT cell death was not caused by the BBB4-G4K NPs but by the BBB4. Note that at the 15 µM concentration critical for cells, the amount of free BBB4 released in the medium corresponded to the concentrations of 173.4, 239.9, and 254.7 µM after 4, 12, and 24 h of exposure, respectively. Such concentrations are 1.7-, 2.4- and 2.5-fold higher than the maximum concentration of BBB4 tested, which after 24 h of exposure killed 60% of the cells. Moreover, at such high concentrations, the BBB4 released tended to precipitate into the medium, forming macro-crystals that disturbed the cells with a non-specific mechanical action, probably damaging irreparably their cytoplasmic membrane. In this regard, Figure 3 shows the macro-crystals formed by BBB4 released in the medium at a concentration of 15 µM.

To visualize the cytotoxicity profiles of the three samples, they are represented with bar graphs in Figure 2. To obtain dispersion graphs, the data of the cell viability (%) at 4, 12, and 24 h were graphed vs. the concentrations tested for the G4K, BBB4, and BBB4-G4K NPs. The curves of the cell’s viability (%), as a function of the compound’s concentrations, were obtained and are available in Appendix A.

Once having obtained all the curves in Appendix A, to understand exactly whether, with its nanotechnological manipulation, the cytotoxicity of the BBB4 had been reduced or improved, and to calculate the *SI* values (*LD*_50_/MIC) of both the BBB4 and the BBB4-G4K NPs, we considered the dispersion graphs of the cell viability% vs. the concentrations of G4K, BBB4, and BBB4-G4K NPs at 24 h of exposure. Notably, we determined the *LD*_50_ of all the samples, using the equations of regression models, which best fit the considered dispersion graphs (Figure 4a,b).

The best fitting regressions were chosen based on the value of the related coefficient of determination R^2^. Accordingly, the regression models of the dispersion graphs of the BBB4 and G4 K were polynomial, while that of the BBB4-G4K NPs was linear. Note that because at concentrations of BBB4-G4K NPs > 15 µM the cell viability remained constant, these data were not considered to obtain for the linear regression model of BBB4-G4K NPs its tendency line and relative equation. Table 4 collects the three equations, the associated R^2^ values, the *LD*_50_, and the range of *SI* for the untreated BBB4, BBB4-G4K NPs and for the nanoengineered BBB4 released by the NPs according to the *LD*_50_ determined for the BBB4-G4K NPs; the *SI* values computed for each staphylococcal isolate are observable in Table 3. 

According to the data in Table 4, from a superficial deduction, it could seem that by its nano-encapsulation in G4K, the BBB4 cytotoxicity on the HaCaT cells was increased, but this is not the case.

Indeed, for a correct interpretation of the results, we must consider the micromolar concentration of BBB4 that the *LD*_50_ of BBB4-G4K NPs can deliver after 24 h in accordance with its DL% (28.8%) and release profile (95.5% cumulative release). Being at such a concentration as 138.6 µM, the *LD*_50_ of the nanotechnologically manipulated BBB4 resulted in being 2.5-fold higher than that of the untreated BBB4, thus establishing that by formulating BBB4 in NPs using G4K, not only was its solubility in water improved but also its cytotoxicity was reduced by 2.5 times.

Furthermore, excluding the staphylococcal strains against which BBB4 displayed MICs > 128 µg/mL and was considered inactive, the *SI* values of the BBB4-G4K NPs, as well as those of the nano-manipulated BBB4, were 2.3–18.0-fold higher than those of the BBB4, thus establishing their greater suitability for potential biomedical applications as a new antibacterial therapeutic agent. To understand if the *SI* values determined for the BBB4-G4K NPs could be satisfactory, we analyzed some reported opinions, which unfortunately were of little help because they conflicted. According to some authors, *SI* values of ≥10 are necessary to make a molecule worthy of further investigation [55,56]. Weerapreeyakul et al. [57] proposed *SI* values of ≥3 to define a clinically applicable molecule as an anti-cancer agent. In microbiology, Adamu et al. [58,59] reported *SI* values of ≤5.2 for South African plant leaf extracts with antibacterial properties. Famuyide et al. [60] stated that antibacterial plant extracts could be considered bioactive and non-toxic if the was *SI* >1, while Nogueira and Estolio do Rosario reported that the *SI* should not be less than 2 [61]. We thus believe that further studies are necessary to clearly determine the minimum acceptable *SI* value. Nevertheless, considering the many divergent opinions on the *SI* acceptance criterion, we trust that the *SI* values determined here for BBB4-G4K NPs could be considered acceptable to suggest BBB4-G4K NPs as a promising antibacterial agent suitable for clinical development. 

## 4. Conclusions

Water-soluble, BBB4-loaded NPs (BBB4-G4K NPs), previously obtained by entrapping BBB4 in a non-bioactive, polyester-based, lysine-containing fourth generation cationic dendrimer (G4K), have demonstrated remarkable antibacterial effects against numerous strains of different species of the *Staphylococcus* genus, including MRSA and MRSE isolates also resistant to linezolid, and other species of staphylococci resistant to oxacillin. The micromolar *MIC* values determined for the BBB4-G4K NPs on 23 staphylococci were very uniform and in the range of 1.5–6 µM, regardless of the drug resistance patterns of the strains tested. Interestingly, the BBB4-G4K NPs have also been able to inhibit strains of *S. epidermidis*, against which even the most recent antibiotics, such as linezolid, are rapidly losing efficacy. In all cases, the MICs of the BBB4-G4K NPs were lower than those determined for the free BBB4, thus confirming that our nanotechnological approach not only succeeded in obtaining a water-soluble BBB4-formulation suitable for future in vivo applications, but also improved the antibacterial potency of the BBB4.

Staphylococci, and particularly *S. aureus*, in addition to being responsible for respiratory infections, such as sinusitis, and food intoxications and for more severe and life-threatening diseases, such as pneumonia, meningitis, osteomyelitis, endocarditis, toxic shock syndrome, and sepsis, frequently cause skin and soft tissue infections, including surgical-wound infections, pimples, impetigo, boils, cellulitis, folliculitis, abscesses, carbuncles, and scalded skin syndrome. Consequently, to evaluate the possible future clinical application of BBB4-G4K NPs as a therapeutic agent, especially for skin infections, we examined its cytotoxicity on human keratinocyte cells (HaCaT).

The results showed that what was actually responsible for the extensive cell death observable after the BBB4-G4K NP administration at a dosage ≥ 15 µM was the BBB4 and not the nano-formulation, due to the high amount of BBB4 released in the medium at that concentration. In particular, the evident formation of BBB4 macro-crystals in the medium probably irreparably damaged the cytoplasmic membrane of the HaCaT cells, causing their death. Overall, our findings established that the *LD*_50_ of the nanotechnologically manipulated BBB4 resulted in being 2.5-fold higher than that of untreated BBB4, thus establishing that by formulating BBB4 in NPs using G4K, not only were its water solubility and its antibacterial potency improved but also its cytotoxicity was significantly reduced.

Additionally, the *SI* values calculated for BBB4-G4K NPs, considering the staphylococci, against which the BBB4 was also active, were in the range of 2.7–5.5 and were 2.3–18.0-fold higher than those of the BBB4, thus establishing their higher suitability for possible biomedical applications as a new antibacterial therapeutic agent. Moreover, the *SI* values of the BBB4-G4K NPs were fully compliant with those reported as acceptable to allow the therapeutic application of a new drug. Finally, as the *SI* values of the nanoengineered BBB4 were identical to those of the BBB4-G4K NPs (2.7–5.5), as with those of the BBB4-G4K NPs, they were 2.3–18-fold higher than those of the untreated BBB4, thus further confirming the effectiveness of the innovative nanotechnological strategy executed for improving the properties of the pyrazole nucleus. 

With our study, we have detected a powerful antibacterial compound which was highly selective for the *Staphylococcus* genus. Overall, the novelty of our research consists in having successfully created a new nanotechnological antibacterial agent based on a pyrazole derivative (BBB4), which in our previous studies was not applicable in vivo in its free form because it was not soluble in water. The acquired solubility in water of the new BBB4 nano-formulation and its selectivity for bacteria cells makes possible its safe administration in vivo without the use of harmful solvents, co-solvents, and surfactants. 

## Figures and Tables

**Figure 1 biomedicines-10-00017-f001:**
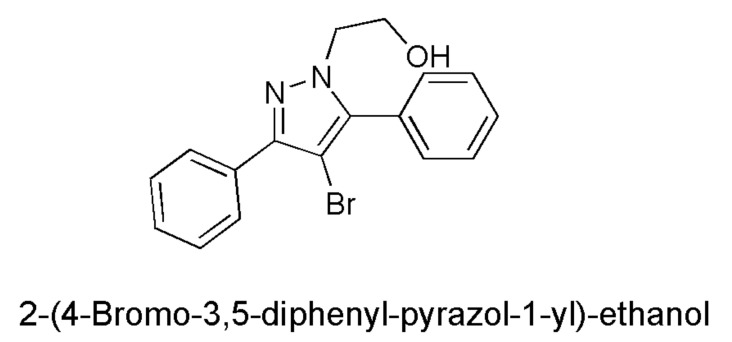
Chemical structure of BBB4.

**Figure 2 biomedicines-10-00017-f002:**
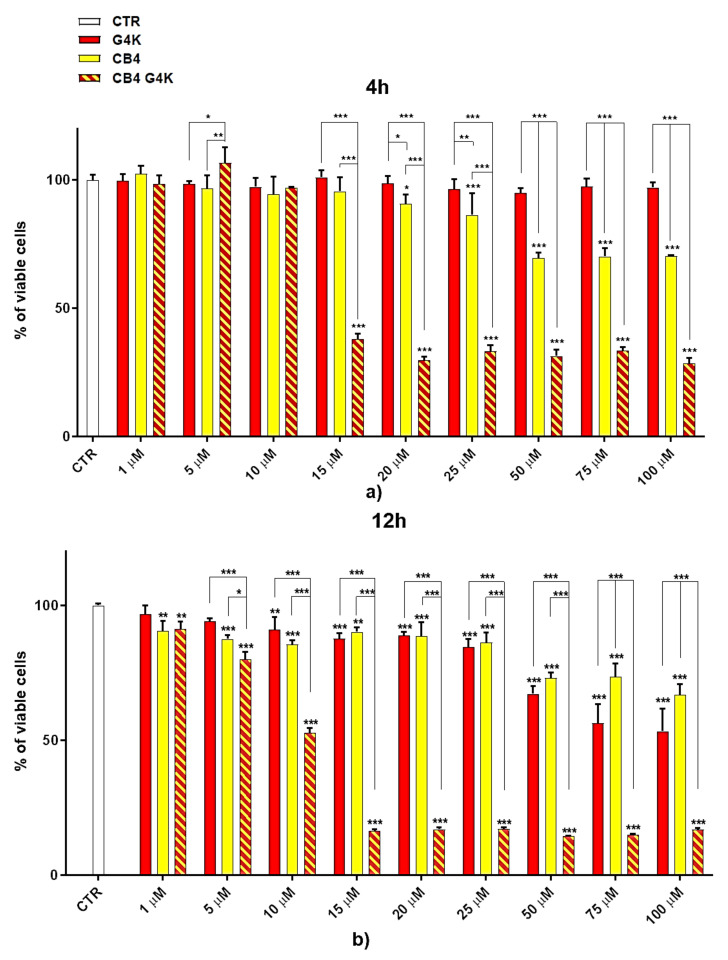
Dose- and time-dependent cytotoxicity activity of G4K, BBB4 (CB4 in the Figure legend), and BBB4-G4K NPs (CB4 G4K in the Figure legend) at 4 h (**a**), 12 h (**b**), and 24 h (**c**) towards HaCaT cells. Where not specified, the significance refers to control (*p* > 0.05 ns; *p* < 0.05 *; *p* < 0.01 **; *p* < 0.001 ***).

**Figure 3 biomedicines-10-00017-f003:**
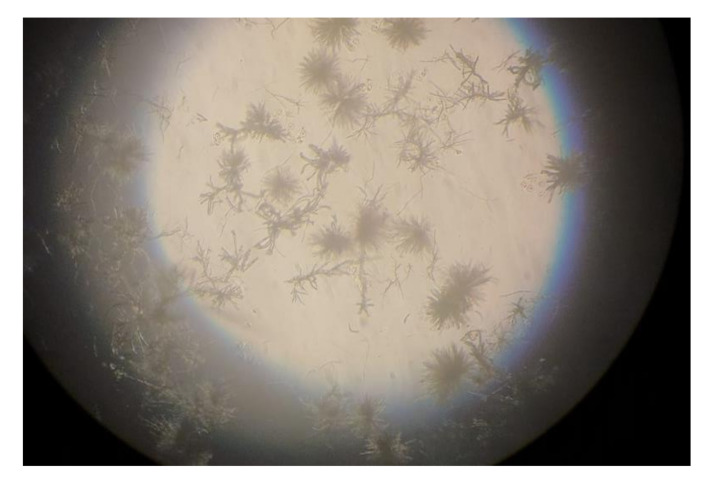
Macro-crystals formed by the free BBB4 released in the medium (173.4 µM after 4 h, 239.9 µM after 12 h, and 254 µM after 24 h) from BBB4-G4K NPs at 15 µM, responsible for the cell death.

**Figure 4 biomedicines-10-00017-f004:**
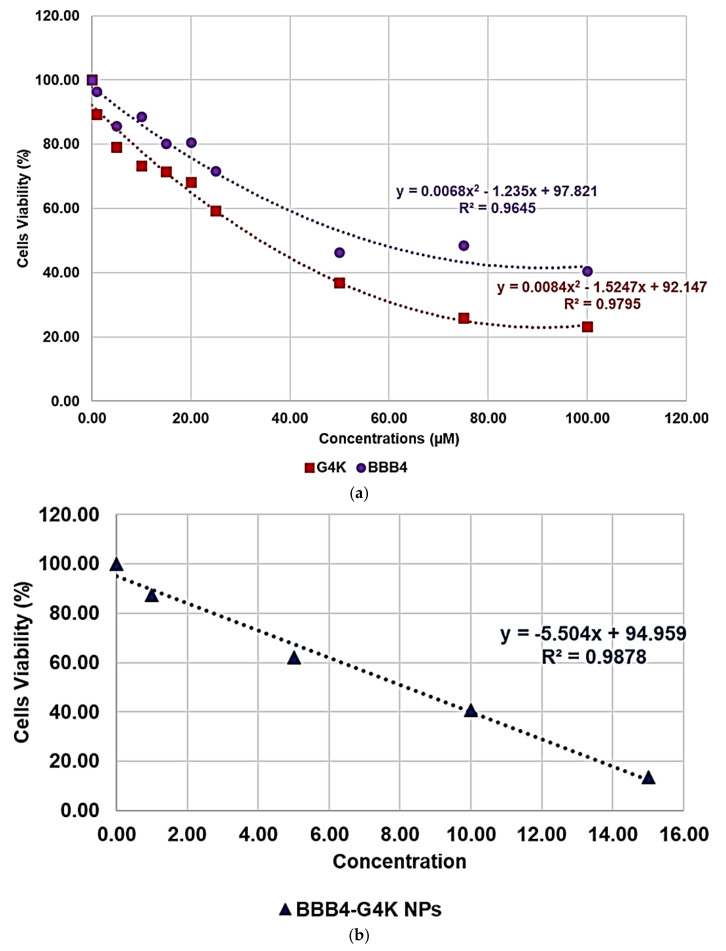
Regression models that better fit the dispersion graphs obtained, reporting in the graph the cell viability% vs. the concentration of samples at 24 h of exposure of BBB4 and G4K (**a**) and of BBB4-G4K NPs (**b**).

**Table 1 biomedicines-10-00017-t001:** Main characteristic of BBB4-G4K NPs [16].

Analysis	BBB4-G4K NPs	
FTIR [cm^−1^]	Bands Attribution 3500–3000 (NH_3_^+^ dendrimer, OH stretching BBB4) 2985, 2880 (alkyl groups of dendrimer and BBB4) 1736 (C=O stretching esters of dendrimer) 1220, 1051 (C-O stretching esters of dendrimer) 697 (C-Br stretching of BBB4)	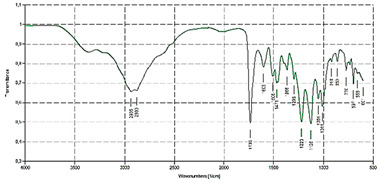
^1^H NMR (400 MHz, CD_3_OD) [ppm]	Signals attribution **1** = 423 H (CH_3_ of G4K + CH_2_CH_2_CH_2_ of lys) **2** = 96 H (CH_2_NH_3_^+^ of lys) **3** = 120H (CH_2_ of BBB4 + CHNH_3_^+^ of lys + CH_2_ of BBB4) **4** = 186 H (CH_2_O of G4K) **5** = 144 H (CH= of phenyl rings) **6** = 36 H (CH= of phenyl rings)	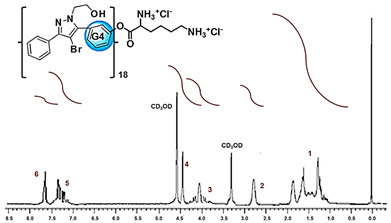
HPLC	DL (%) EE (%)	28.8 ± 1.2 39.0 ± 1.6
^1^H NMR	MW	21,175.8
DL% by HPLC		21,072.6 ± 240.2
Scanning Electron Microscopy (SEM)	Morphology Average Size	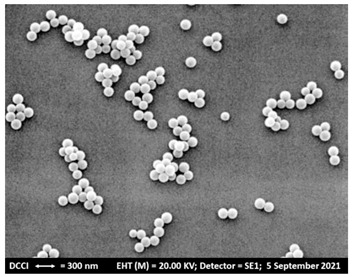
DLS ^1^ Analysis	Z-Ave ^2,5^ (nm) PDI ^3,5^	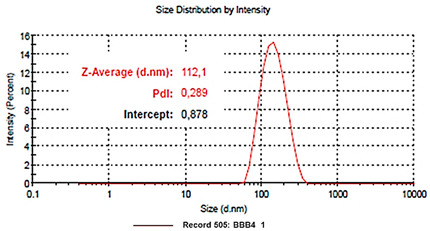
	ζ-p ^4,5^ (mV)	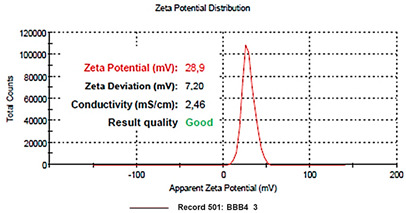
Solubilization Essay ^§^	Water-Solubility (mg/mL)	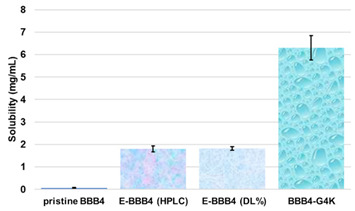
Dialysis Method (HPLC)	Cumulative Release (%, 24 h)	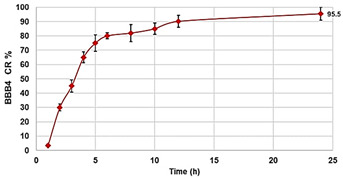
	Mathematical Model	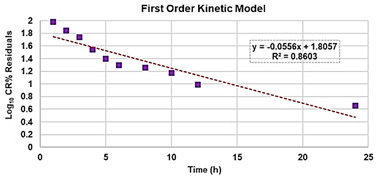
	Mechanism	Drug release rate is concentration-dependent
G4K Cytotoxicity (HeLa Cells)	Cell Viability (%) (0–100 µM)	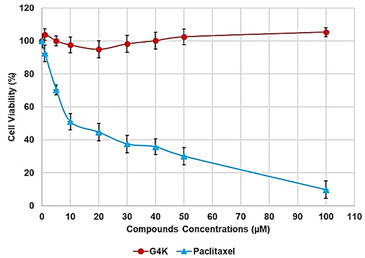
Potentiometric Titration ^#^	Protonation Profile	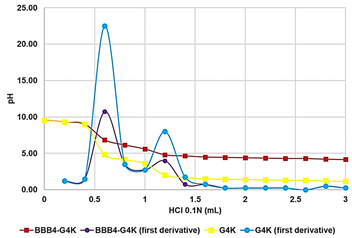

^1^ dynamic light scattering; ^2^ hydrodynamic diameters of particles; ^3^ polydispersity indices; ^4^ measure of the electrical charge of particles suspended in the liquid of acquisition (water); ^5^ correspondent values for G4K = 333.4 ± 24.6, 0.286 ± 0.040, +66.1 ± 4.7; ^§^ the image also shows the water solubility of untreated BBB4, nanoengineered BBB4 released in water, and BBB4 contained in NPs; Paclitaxel = positive control; ^#^ the image also shows the titration curve and the first derivative of G4K.

**Table 2 biomedicines-10-00017-t002:** MICs of BBB4 as µg/mL and micromolarity (µM), against representative clinical isolates of Gram-positive and Gram-negative species, including several strains of the *Staphylococcus* genus. The reference strain *S. aureus* ATCC 29213 was also employed.

	BBB4 MW 343.2		Reference Antibiotics	
	*MIC* Values ^$^		*MIC* Values ^$^	
	µg/mL	µM	µg/mL	µM
Gram-negative enterobacteriaceae and non-fermenting bacteria				
*E. coli* 224 ^#^	>128	>373	8 32 16	17 ^1^ 97 ^2^ 27 ^3^
*P. aeruginosa* 247	>128	>373	64	78 ^4^
Gram-positive sporogenic				
*B. subtilis* 394	128	373	128	212 ^3^
Gram-positive staphylococci				
*S. aureus* ATCC 29213	32	93	<2 ^5^	<5 ^5^
*S. aureus* 18 *	32	93	512 ^5^	1275 ^5^
*S. aureus* 191 *	32	93	512 ^5^	1275 ^5^
*S. aureus* 195 *	32	93	256 ^5^	638 ^5^
*S. epidermidis* 22 ^§^	32	93	256 ^5^	638 ^5^
*S. epidermidis* 197 ^§^	32	93	256 ^5^	638 ^5^
S. epidermidis 198 ^†,§^	32	93	256 ^5^	638 ^5^
*S. epidermidis* 181 ^†,§^	32	93	256 ^5^	638 ^5^
*S. haemoliticus* 193 ^††^	>128	>373	256 ^5^	638 ^5^
*S. hominis* 125 ^††^	>128	>372	256 ^5^	638 ^5^
*S. lugdunensis* 129	128	373	<2 ^5^	<5 ^5^
*S. saprophyticus* 41	32	93	<2 ^5^	<5 ^5^
*S. symulans* 163 ^††^	>128	>373	256 ^5^	638 ^5^
*S. warneri* 74 ^††^	64	187	256 ^5^	638 ^5^
Gram-positive enterococci				
*E. faecium* 341 ^#^	>128	>373	256	177 ^6^
*E. faecalis* 1 ^#^	>128	>373	128	88 ^6^

^$^ Experiments were performed in triplicate, the concordance degree was 3/3, and ±SD was zero; ^#^ indicates antibiotics susceptibility; *P. aeruginosa* was MDR; * indicates MRSA; ^§^ indicates MRSE; ^†^ indicates MRSE also resistant to linezolid; ^#^ indicates VRE; ^††^ indicates resistance to oxacillin; ^1^ ertapenem; ^2^ ciprofloxacin; ^3^ amoxy-clavulanate; ^4^ piperacillin taxobactam; ^5^ oxacillin; ^6^ vancomycin.

**Table 3 biomedicines-10-00017-t003:** MICs of BBB4-G4K NPs, expressed as µg/mL and micromolarity (µM), against the *Staphylococcus* species tested in the study; the relative maximum concentrations of BBB4 that should be released by BBB4-G4 NPs according to the observed MICs; and the relative DL% and the release profile [16]. The MICs of pristine BBB4 were also reported. The last two columns report the values of selectivity indices (*SI*) of BBB4 and BBB4-G4K NPs.

	BBB4 MW 343.2	BBB4-G4K MW 21,175.8	Max BBB4 Releasable ^1^	Oxacillin	*SI*^2^ BBB4	*SI*^2^ BBB4-G4K
*MIC* Values ^3^	µg/mL, µM	µg/mL, µM	µg/mL, µM	µg/mL, µM		
*S. aureus* 17 *	32, 93.2	64, 3.0	17.6, 51.3	512, 1275.5	0.6	2.7
*S. aureus* 18 *	16, 46.6	64, 3.0	17.6, 51.3	512, 1275.5	1.2	2.7
*S. aureus* 191 *	32, 93.2	64, 3.0	17.6, 51.3	512, 1275.5	0.6	2.7
*S. aureus* 195 *	32, 93.2	32, 1.5	8.8, 25.7	256, 637.8	0.6	5.5
*S. aureus* 292 *	32, 93.2	64, 3.0	17.6, 51.3	512, 1275.5	0.6	2.7
*S. aureus* 293 ***	32, 93.2	64, 3.0	17.6, 51.3	512, 1275.5	0.6	2.7
*S. aureus* 414	16, 46.6	32, 1.5	8.8, 25.7	<2.0, <5.0	1.2	5.5
*S. aureus* 187 *	16, 46.6	32, 1.5	8.8, 25.7	256, 637.8	1.2	5.5
*S. aureus* 188 *	32, 93.2	64, 3.0	17.6, 51.3	512, 1275.5	0.6	2.7
*S. aureus* 26 *	32, 93.2	64, 3.0	17.6, 51.3	512, 1275.5	0.6	2.7
*S. epidermidis* 22 ^§^	32, 93.2	64, 3.0	17.6, 51.3	512, 1275.5	0.6	2.7
*S. epidermidis* 180 ^§^	32, 93.2	64, 3.0	17.6, 51.3	512, 1275.5	0.6	2.7
*S. epidermidis* 181 ^†,§^	32, 93.2	64, 3.0	17.6, 51.3	512, 1275.5	0.6	2.7
*S. epidermidis* 197 ^§^	32, 93.2	64, 3.0	17.6, 51.3	512, 1275.5	0.6	2.7
*S. epidermidis* 198 ^†,§^	32,93.2	64, 3.0	17.6, 51.3	512, 1275.5	0.6	2.7
*S. epidermidis* 222 ^§^	32, 93.2	64, 3.0	17.6, 51.3	512, 1275.5	0.6.	2.7
*S. haemoliticus* 193 ^††^	>128, >373.0	128, 6.0	35.2, 102.6	512, 1275.5	<0.15	1.4
*S. hominis* 125 ^††^	>128, >373.0	128, 6.0	35.2, 102.6	512, 1275.5	<0.15	1.4
*S. lugdunensis* 129	128, 373.0	64, 3.0	17.6, 51.3	<2.0, <5.0	0.15	2.7
*S. saprophyticus* 41	32, 93.2	64, 3.0	17.6, 51.3	<2.0, <5.0	0.6	2.7
*S. symulans* 163 ^††^	>128, >373.0	128, 6.0	35.2, 102.6	512, 1275.5	<0.15	1.4
*S. warneri* 74 ^††^	64, 186.5	64, 3.0	17.6, 51.3	256, 637.8	0.3	2.7

^1^ Derived from the MICs observed for BBB4-G4K NPs after 24 h and according to their DL% and the BBB4 released from BBB4-G4K NPs after 24 h [16]; ^2^
*SI* = selectivity index (*LD*_50_/MIC) where *LD*_50_ was calculated using data from cytotoxicity experiments on human keratinocytes after 24 h exposure; ^3^ experiments were performed in triplicate, the concordance degree was 3/3, and ±SD was zero; * indicates MRSA; ^§^ indicates MRSE; ^†^ indicates MRSE also resistant to linezolid; ^††^ indicates resistance to oxacillin.

**Table 4 biomedicines-10-00017-t004:** Regression’s equations, R^2^ values, and *LD*_50_ of G4K, BBB4, and BBB4-G4K NPs and of the nanoengineered BBB4 released by NPs according to the *LD*_50_ determined for BBB4-G4K NPs (24 h treatments). The last column reports the *SI* ranges of pristine BBB4 and BBB4-G4K NPs and of the nanoengineered BBB4 released by NPs according to the *LD*_50_ determined for BBB4-G4K NPs (24 h treatments).

Sample	Equations	R^2^	*LD*_50_ (µM)	SI
G4K	y = 0.0084x^2^ − 1.5247x + 92.147	0.9795	34.01	N.A.
BBB4	y = 0.0068x^2^ − 1.235x + 97.821	0.9645	55.97	0.15–1.2 ^1^
BBB4-G4K NPs	y = −5.504x + 94.959	0.9878	8.17	1.4–5.5 ^2^
BBB4 released *	N.A.	N.A.	138.6	1.4–5.5 ^2^

* Means the BBB4 nanotechnologically modified using G4K and released in the medium after 24 h in accordance with the DL% and release profile of BBB4-G4K NPs (see also Appendix A) [16]; N.A. = not applicable; ^1^ computed excluding the staphylococcal strains against which BBB4 displayed MICs > 128 µg/mL; ^2^ computed considering all staphylococcal strains tested.

## Data Availability

All data necessary to support reported results are present in the main text of the article and in the Appendix A.

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
