# Peer review of "Pyrazole-Based Water-Soluble Dendrimer Nanoparticles as a Potential New Agent against Staphylococci"

_biomedicines, 2021, doi:10.3390/biomedicines10010017_

Round 1
Reviewer 1 Report
The article 'Pyrazole-Based Water-Soluble Dendrimer Nanoparticles as a Potential New Agent Against Staphylococci' describes the study of water-suluble dendrimer nanoparticles, loaded with 2-(4-bromo-3,5-diphenyl-pyrazol-1-yl)-ethanol (BBB4), against isolates of S. aureus and S. epidermidis. These nanoparticles demonstrated higher efficiency and lower citotoxicity in comparison with BBB4, and can be considered as promising medicinal agent against staphylococcal skin infections. The results of this study seem new and significant, the manuscript meets aims, scope and high level of Biomedicines journal, and can be accepted after major (still technical) revision.
- The main comment relates to duplication of the data reported previosly (ref. [16]) in Section 3.1. This Section should be removed from the manuscript, the synthesis and sharacterization of BBB4-G4K NPs can be briefly described in Section 2 with reference to [16] and Supplementary Materials.
- The comparative study of the antimicrobal activity of BBB4 (Table 2) BBB4-G4K (Table 3) and antibiotics: for different bacterial cultures MIC values were equal, or determined in wide range. More details are needed, including experimental (the method described in Section 2.2.2 should give MIC values, not intervals).
- The Section 3.2.2 seems more appropriate in Introduction.
- Discussions and argument of really low toxicity of BBB4-G4K are too wordy, the data on Figure 4 duplicates the data presented in Figure 2. Please shorten and clarify the Section 3.3.1.
Minor remarks
Template for Pharmaceutics journal was used for pages 1–9. Please correct.
Line 13 – 'Correspondence:' twice in a row, please correct
Lines 51-52 – 'penicillin-resistant Streptococcus pneumoniae, resistant to penicillin', please remove 'resistant to penicillin'
Line 52 – 'vancomycin-resistant' seems preferable.
etc.
The manusctipt contains a large number of minor bags, and therefore needs substantial editing.
Author Response
The article 'Pyrazole-Based Water-Soluble Dendrimer Nanoparticles as a Potential New Agent Against Staphylococci' describes the study of water-soluble dendrimer nanoparticles, loaded with 2-(4-bromo-3,5-diphenyl-pyrazol-1-yl)-ethanol (BBB4), against isolates of S. aureus and S. epidermidis. These nanoparticles demonstrated higher efficiency and lower cytotoxicity in comparison with BBB4 and can be considered as promising medicinal agent against staphylococcal skin infections. The results of this study seem new and significant, the manuscript meets aims, scope and high level of Biomedicines journal, and can be accepted after major (still technical) revision.
1. The main comment relates to duplication of the data reported previously (ref. [16]) in Section 3.1. This Section should be removed from the manuscript, the synthesis and characterization of BBB4-G4K NPs can be briefly described in Section 2 with reference to [16] and Supplementary Materials.
We agree with the Reviewer concerning the text reported in Section 3.1. and thank him for his suggestion to lighten that part. Accordingly, practically all the text of Section 3.1 has been removed and title has been modified (lines 268-291). Table 1 has been maintained because, in our opinion, it can be useful for readers to have at hand the main characteristics of the BBB4-G4K nanoparticles (NPs) that, previously synthetized and characterized, have been subjected to biological tests in this work. We therefore kindly ask the Reviewer to allow us to leave Table 1 in our manuscript.
2. The comparative study of the antimicrobial activity of BBB4 (Table 2) BBB4-G4K (Table 3) and antibiotics: for different bacterial cultures MIC values were equal, or determined in wide range. More details are needed, including experimental (the method described in Section 2.2.2 should give MIC values, not intervals).
We absolutely agree with what the Reviewer highlighted.
Tables 2 and 3 now shows the individual MIC values ​​for the different reference antibiotics calculated for each strain.
3. The Section 3.2.2 seems more appropriate in Introduction.
We thank the Reviewer for his suggestion that we have put into practice. Particularly, the original Section 3.2.2 (lines 403-455) has been moved to the Introduction as subsection 1.2.1. of Section 1.2. Please check the lines 146-197. Consequently, the numbers of some references have been changed in the text (original Refs. 43-53 now are Refs. 40-50 and Refs. 40-42, now are Refs. 51-53), and the list of references has also been updated.
4. Discussions and argument of really low toxicity of BBB4-G4K are too wordy, the data on Figure 4 duplicates the data presented in Figure 2. Please shorten and clarify the Section 3.3.1.
As requested by the Reviewer, Section 3.3.1. has been lightened by eliminating too verbose phrases and repeated concepts. Please, see the parts deleted in Section 3.3.1. Regarding Figure 4, we agree with the Reviewer that it duplicates the data reported in Figure 2, but differently from Figure 2, it reports other types of graphs (dispersion graphs), which are indispensable for determining the equations of the tendency lines relating to the dispersion graphs of G4K, BBB4 and BBB4-G4K NPs (24h), necessary to calculate their LD50. However, to please the Reviewer, explanations concerning this question have been added in the main text (lines 554-559 and 569-576) and Figure 4 has been moved to Supplementary Materials in which it appears as Figure S7 in section S2.2. Original Figure 5 has been updated at Figure 4.
Minor remarks
Template for Pharmaceutics journal was used for pages 1–9. Please correct.
Corrected.
Line 13 – 'Correspondence:' twice in a row, please correct
Corrected.
Lines 51-52 – 'penicillin-resistant Streptococcus pneumoniae, resistant to penicillin', please remove 'resistant to penicillin'
Removed
Line 52 – 'vancomycin-resistant' seems preferable.
Vancomycin-resistant has been used as suggested by the Reviewer (line 52).
The manuscript contains a large number of minor bags, and therefore needs substantial editing.
We apologize to the Reviewer for our inattentions. We thoroughly checked the entire manuscript and cleaned it of all typos we found (corrections highlighted in different colours).
Reviewer 2 Report
The authors synthesized Pyrazole-Based Water-Soluble Dendrimer Nanoparticles as anti-bacterial agents. The title seems interesting. I have a few concerns that must be addressed before publication.
-Why the anti-bacterial effects of the prepared nanoformulation were tested on different strains of different species of the Staphylococcus genus? Did the authors select a standard strain as well?
- How did the authors provide BBB4? Is it purchased?
- The authors need to elaborate on the biomedical application (such as anti-bacterial- and anti-cancer-effects) of nanoparticles by citing and discussing the following papers;
DOI: 10.3390/nano11102535, 10.1007/s00339-021-04917-8, 10.3390/ma14040825
- Why did the authors choose HaCaT cells for cytotoxicity assays? IC50 values should also be calculated for each incubation time. Also, investigating cells viability and drug release after 48 h seems to be helpful to draw a reliable conclusion.
- What are the limitations of the study?
Author Response
The authors synthesized Pyrazole-Based Water-Soluble Dendrimer Nanoparticles as anti-bacterial agents. The title seems interesting. I have a few concerns that must be addressed before publication.
-Why the anti-bacterial effects of the prepared nanoformulation were tested on different strains of different species of the Staphylococcus genus? Did the authors select a standard strain as well?
We thank the Reviewer for his deep interest in the rational of the experiments performed in our study. We explain that, as clearly reported in Section 3.2, we firstly examined the antibacterial activity of BBB4 on strains, also MDR, representative of species both Gram-positive and Gram-negative, including an S. aureus ATCC strain, different species of the Staphylococcus genus, isolates of the Enterococcus genus, and isolates of E. coli, Pseudomonas aeruginosa and Bacillus subtilis. From the obtained results, BBB4 was considered inactive against isolates of Gram-negative species and of the Enterococcus genus, poorly active against B. subtilis and promising against some isolates of the Staphylococcus genus, and mainly towards strains of the species S. aureus and S. epidermidis.
Consequently, we thought it interesting to deepen the study of the antibacterial activity of BBB4 against a wider range of isolates of the Staphylococcus genus. On the other hand, to obtain not questionable MIC values, due to the poor solubility of BBB4, we have recently encapsulated BBB4 in the G4K dendrimer, proven to be devoid of antibacterial properties (results not reported), obtaining water-soluble BBB4-loaded NPs with also physicochemical properties suitable for a future biomedical application. In the herein study, we evaluated whether, after this nano-manipulation, the antibacterial properties of BBB4 already observed had been preserved and/or modified. Please, see lines 305-314 and lines 322-336.
- How did the authors provide BBB4? Is it purchased?
As the Reviewer can infer from the Introduction and by consulting reference 16, BBB4 was synthesized in 1998 by Professor Bondavalli and his collaborators who tested it for different biological activities, except for the antibacterial one. However, this has now been better specified in the main text (line 78).
- The authors need to elaborate on the biomedical application (such as anti-bacterial- and anti-cancer-effects) of nanoparticles by citing and discussing the following papers;
DOI: 10.3390/nano11102535, 10.1007/s00339-021-04917-8, 10.3390/ma14040825
We apologize in advance to the Reviewer, but we believe that even a minimal discussion on the biomedical application of nanoparticles as anti-cancer and/or antibacterial agents is not appropriate for the present study.
In fact, the purpose of this study was to evaluate and discuss the antibacterial activity and cytotoxicity of the nano-formulation we prepared. In this contest, a general discussion on nanoparticles applicability would be out of scope. Consequently, the Reviewer's request to cite the three proposed works cannot be satisfied. It could have been satisfied in our previous work (Ref. 16) recently published in Nanomaterial where the main topic was the preparation and characterization of water-soluble nanoparticles with potential for future clinical uses, but unfortunately for the Reviewer such work is already published. We therefore ask the Reviewer to understand our position.
- Why did the authors choose HaCaT cells for cytotoxicity assays?
To answer to the request of the Reviewer we cite a significant part of the manuscript in which the Reviewer can find the due explanation.
“Such experiments were performed on HaCaT keratinocytes mainly because the 75 % of MRSA infections are localized to the skin and soft tissues [31,43]. Consequently, to assess the cytotoxicity of BBB4-G4K NPs, we selected human keratinocytes, which are the primary type of cell found in the epidermis, the outermost layer of the skin, and are more susceptible to colonization by bacteria, fungi, and parasites.”
IC50 values should also be calculated for each incubation time.
We make kindly note to the Reviewer that, concerning BBB4 and G4K NPs, calculating the LD50 after 4 and 12 hours of exposure was impossible because the viability of the cells remained above 50% even at the highest concentration considered. On the other hand, as regards BBB4-G4K NPs, the LD50 values at 4 and 12 hours of exposure were not significantly different from the value determined for 24 hours of exposure. Consequently, given that we were interested in evaluating the LD50 to calculate the selectivity index which is given by LD50/MICs, where the MIC is determined after 24 h of exposure, we reported and used the LD50 data at 24 hours.
Also, investigating cells viability and drug release after 48 h seems to be helpful to draw a reliable conclusion.
Similarly, both cytotoxicity and releases were determined after 24 hours to obtain data comparable to the MICs. Furthermore, given that after 24 hours, the release of the drug was almost quantitative (96%), it would not have made sense to go beyond 24 hours.
- What are the limitations of the study?
We have repeatedly considered the design of the experiments and the results obtained in our study and they have always appeared consequential and very promising. We are therefore confident that we can state that this study has no particular limitations.
Round 2
Reviewer 1 Report
The authors substantially revorked the manuscript taking into account the comments of the Reviewers. As presented, the article can be accepted for publication.